# Prevalence and factors associated with underweight among children under five born to adolescent mothers: Evidence from the 2022 Kenya Demographic and Health Survey

**Alfred Omutoj[1], Henry Prosper Dade [2]\*, Annitah Kagali[1], Samuel Salu[3], David Mensah Otoo [3], Betty Oloo[1], Prince Tsekpetse[1]**

**1** Department of Community and Public Health, Busitema University, Mbale, Uganda, **2** Nurses' Training College, Pantang, Ghana, **3** Department of Epidemiology and Biostatistics, University of Health and Allied Sciences, Hohoe, Ghana

\* henrydadep@gmail.com

## Abstract

### Background

Undernutrition remains a major public health concern in low- and middle-income countries, with children born to adolescent mothers being particularly at risk of underweight. In Kenya, despite high adolescent birth rates, there is limited nationally representative evidence on the prevalence of underweight and its determinants among children under five years born to adolescent mothers. This study examined the prevalence and factors associated with underweight among Kenyan children under five years of age born to adolescent mothers, using nationally representative data from the 2022 Kenya Demographic and Health Survey (KDHS).

### Methods

This study was a cross-sectional analysis of data from the child recode file (KR) of the 2022 KDHS, comprising a weighted sample of 819 children under five years born to adolescent mothers aged 15–19 years. Underweight was defined as a weight-for-age Z-score of less than −2 SD. Modified Poisson regression analyses were performed to identify factors associated with underweight. All analyses accounted for the complex survey design and sampling weights. Results from the bivariate model are presented as crude prevalence ratios (CPR), while results from the multivariable model were presented as adjusted prevalence ratios (APR). Statistical significance was set at P < 0.05.

### Results

The prevalence of underweight among Kenyan children born to adolescent mothers was 11.03% (n = 77/702; 95% CI: 8.65–13.97). In the multivariable model, children of married adolescent mothers were more likely to be underweight (APR = 2.22; 95%

**Data availability statement:** The data used for this study are publicly available from the Demographic and Health Surveys (DHS) Program at https://dhsprogram.com/data/. Access to the data requires free registration with the DHS Program and compliance with their data use agreement.

**Funding:** The author(s) received no specific funding for this work.

**Competing interests:** The authors have declared that no competing interests exist.

**Abbreviations:** KDHS, Kenya Demographic and Health Survey; DHS, Demographic and Health Survey; WHO, World Health Organisation; AOR, Adjusted Odds Ratio; COR, Crude Odds Ratio; SD, Standard Deviation; STROBE, Strengthening the Reporting of Observational Studies in Epidemiology

CI: 1.15–4.28; P = 0.018) than those of unmarried mothers. Additionally, the prevalence of underweight increased with an increase in the child's age. Children aged 6–23 months (APR = 2.73; 95% CI: 1.22–6.12; P = 0.015) and those aged ≥24 months (APR = 4.35; 95% CI: 1.84–10.30; P = 0.001) were more likely to be underweight than those aged six months and below.

## Conclusion

Approximately 1 in 9 children under five years born to adolescent mothers in Kenya were underweight. Being in a marital union and an increase in the child's age emerged as key factors associated with underweight among children born to adolescent mothers. These findings highlight the need for targeted interventions to prevent early marriage among adolescent girls and to improvee complementary feeding practices among adolescent mothers to reduce the risk of underweight among their children..

## Introduction

Undernutrition among children under five remains a critical public health challenge globally, particularly in low- and middle-income countries (LMICs) [1,2]. It manifests in various forms, including stunting, wasting, and underweight [3]. Undernutrition contributes to both immediate and long-term impairments in child survival, growth, and neurodevelopment [4]. According to the World Health Organisation, approximately 150 million children under five were stunted, and 43 million were wasted globally in 2024 [5]. In Africa, approximately 216 million children suffer from undernutrition [6]. Recent estimates from the Kenya Demographic and Health Survey (KDHS) indicate that 3% of children under five years are underweight, 18% are stunted, and 5% are wasted [7].

Adolescent pregnancy, which refers to pregnancy occurring in girls aged 10–19 years, is increasingly recognised as a risk factor for poor maternal and child health outcomes [8,9]. Although adolescent birth rates have declined globally as of 2019, sub-Saharan Africa (SSA) continues to report more than 100 births per 1,000 women, with an estimated 6,114,000 births occurring among girls aged 15–19 years [10]. In Kenya, for instance, 15% of adolescent girls aged 15–19 years have begun childbearing [7]. Adolescent mothers are often socioeconomically disadvantaged, more likely to have limited education, and face barriers in accessing healthcare and adequate nutrition [9]. These vulnerabilities, coupled with biological immaturity, increase the likelihood of adverse birth outcomes, including low birth weight and preterm delivery, which are strongly associated with poor early childhood growth trajectories [11].

Empirical evidence suggests that children born to adolescent mothers are at greater risk of undernutrition, particularly underweight, than those born to older mothers [11–13]. A recent meta-analysis by Welch et al. (2024) found that children of adolescent mothers had significantly higher odds of being moderately and severely underweight than their peers born to adult mothers [9]. However, the findings are

heterogeneous across regions and outcomes; for instance, no statistically significant association was observed between adolescent pregnancy and child wasting in the same review. Similarly, Yu et al. (2016) found regional and age-specific variations, with child height-for-age deficits being most pronounced in the first 12 months and persisting beyond 24 months in many LMICs [11]. These disparities may reflect differences in maternal nutritional status, caregiving behaviours, household food insecurity, and access to maternal-child health services.

In Kenya, several studies have examined the factors associated with undernutrition using small sample sizes, which limits their generalizability [14–16]. Furthermore, there remains a paucity of nationally representative, disaggregated analyses exploring nutritional outcomes among children born to adolescent mothers, a subgroup that is often overlooked in public health research despite being at an elevated risk of undernutrition. This study aims to fill this gap by examining the prevalence and factors associated with underweight among Kenyan children under five years born to adolescent mothers using nationally representative data. Identifying these factors is essential for informing targeted policies and programmatic interventions tailored to this high-risk group. The findings of this study have the potential to guide stakeholders, including the Ministry of Health, adolescent reproductive health programs, and international agencies such as UNICEF, in designing and implementing effective strategies to reduce child undernutrition and break the intergenerational cycle of poverty and malnutrition.

## Methods

### Data source and study design

This study was a cross-sectional analysis of data from the child recode (KR) file of the 2022 Kenya Demographic and Health Survey (KDHS). The Kenya National Bureau of Statistics implemented the 2022 KDHS in collaboration with the Ministry of Health, the National Council for Population and Development, and other national institutions, with technical assistance from ICF through the DHS Program [17].

The KDHS employed a cross-sectional design to gather extensive health and demographic data encompassing a broad range of indicators, including fertility trends, marital patterns, contraceptive use, infant feeding behaviours, nutritional status, public awareness, and attitudes toward HIV/AIDS and other STIs [17]. In keeping with the standards of the International Demographic and Health Survey program, the KDHS employed a nationally representative, stratified, two-stage cluster sampling design, ensuring robust coverage across geographic and sociodemographic strata [17]. In the first stage, 1,692 enumeration areas (EAs) were selected using probability proportional to size sampling. In the second stage, 25,755 households were systematically selected from household listings within these clusters across 47 counties in Kenya. All eligible women aged 15–49 years residing in or staying in the selected households were invited to participate in the study.

We obtained access to the dataset through a formal request approved by ICF International via the DHS Program's online portal (http://www.dhsprogram.com). In preparing this manuscript, the authors followed the Strengthening the Reporting of Observational Studies in Epidemiology (STROBE) reporting framework to ensure methodological transparency and rigour [18] (Fig 1).

The flowchart illustrates the inclusion and exclusion criteria used to derive the final weighted sample of 819 children under five years born to adolescent mothers aged 15–19 years.

### Ethics approval and consent to participate

This study used secondary data from the 2022 Kenya Demographic and Health Survey (KDHS), which are publicly available and fully anonymised. As the analysis involved no direct contact with participants and no access to identifiable information, additional ethical approval was not required. Permission to access and use the KDHS dataset was obtained through a formal request to the DHS Program. All analyses were conducted in accordance with the DHS Program's data use agreement and ethical guidelines for secondary data analysis.

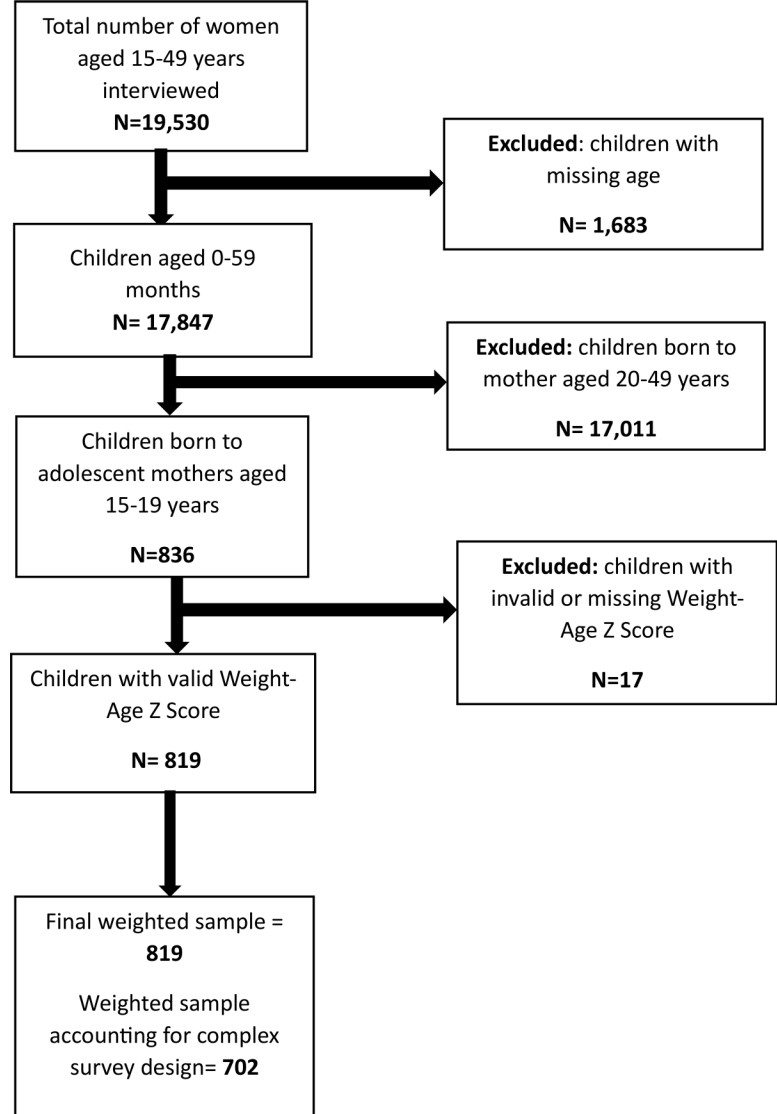

**Fig 1. Flowchart of sample selection and exclusion criteria from the child recode (KR) file of the 2022 KDHS.**

## Study variables

**Outcome variable.** The outcome variable for this study was underweight among children under five years of adolescent mothers. This was assessed by the GDHS in accordance with the World Health Organization (WHO) growth standards. Underweight was defined as a weight-for-age Z-score (WAZ) less than minus two standard deviations (−2 SD) from the median of the WHO reference population.

Weight-for-age is a composite anthropometric index that reflects both acute and chronic undernutrition, incorporating aspects of both stunting (height-for-age) and wasting (weight-for-height). Children with WAZ between −2 SD and −3 SD were categorized as moderately underweight, while those with WAZ below −3 SD were considered severely underweight. Children with WAZ ≥ −2 SD were classified as having normal weight.

For this study, underweight was coded as a binary variable:

$$1 = \text{underweight (WAZ} < -2 \text{ SD)}$$

$$0 = \text{normal weight (WAZ} \geq -2 \text{ SD)}$$

**Covariates.** After an in-depth review of existing literature [12,19–21] and the availability of variables in the dataset, a total of 14 variables were selected as covariates. This included mother's educational level (no formal education, primary, secondary or higher), marital status (not married, married, cohabiting, previously married), wealth index (poorest, poorer, middle, richer, richest), religion (christians, muslim, atheists/other), place of residence (urban, rural), currently working (no, yes), number of household members (<5, ≥5), mothers age at first birth [11–19], child's sex (male, female), child's age-months (<6, 6-, 23, ≥24), parity (1, 2-3), birth order (1, 2-3), fever & cough in the last 2 weeks (no, yes), diarrhea in the last 2 weeks (no, yes).

The wealth index is a standard DHS-derived composite measure constructed using principal component analysis (PCA) of household ownership of assets (e.g., television, bicycle), housing materials, sanitation facilities and water source. Households are ranked and categorized into quintiles (poorest, poorer, middle, richer, richest) by DHS. We used this pre-generated variable without modification.

## Statistical analysis

All statistical analyses were performed using Stata version 17 (StataCorp, College Station, TX, USA). Prior to analysis, observations with missing data or flagged anthropometric measurements were excluded, resulting in a final sample size of 819 participants. All descriptive statistics used the unweighted sample (N = 819), while prevalence estimates and regression analyses were weighted, corresponding to a weighted sample of 702 children. The Stata survey design command (svyset) was used to account for complex survey design.

Descriptive statistics were conducted, and the results were summarised using means, frequencies, and percentages. We used Modified Poisson regression with robust standard errors to examine the association between independent variables and underweight. This method was chosen because the prevalence of underweight exceeded 10%, making odds ratios from logistic regression prone to overestimating the risk. A bivariate model was first fitted, and independent variables with a p-value less than 0.20 were included in the multivariable analysis to avoid excluding potentially important predictors. This threshold allows the inclusion of variables that may not reach conventional levels of significance in the bivariate analysis but could be significant in the multivariable context due to confounding or interaction [22]. Additionally, variables previously reported in the literature to be associated with underweight were included to account for confounding effects [21,23,24]. Results from the bivariate and multivariate analyses are presented as crude and adjusted prevalence ratios (CPR and APR), each with corresponding 95% confidence intervals. Multicollinearity among independent variables was assessed using variance inflation factors (VIFs). Categorical variables were dummy-coded, and VIFs were examined at the level of individual predictors using a linear regression framework. Following standard practice, VIF values greater than 5 were considered indicative of moderate multicollinearity, while values above 10 suggest serious multicollinearity [25]. In this study, all VIF values were below 5, indicating no evidence of problematic multicollinearity.

Statistical significance was set at $P < 0.05$.

## Results

### Background characteristics of the respondent

Table 1 presents the background characteristics of adolescent mothers (n = 819) and their children under five years of age included in the study. The majority of mothers were aged 18–19 years (n = 614, 75.01%), had attained secondary or higher education (n = 394, 48.13%), and were unmarried (n = 373, 45.52%). A significant proportion were from the poorest wealth

**Table 1. Characteristics of the study participants (Weighted N = 819).**

| Variable | Weighted Frequency (N) | Weighted Percentage (%) |
|---|---|---|
| **Mother's age** | | |
| 15-17 | 205 | 24.99 |
| 18-19 | 614 | 75.01 |
| **Mother's educational level** | | |
| No formal education | 55 | 6.69 |
| Primary | 370 | 45.18 |
| Secondary or higher | 394 | 48.13 |
| **Marital status** | | |
| Not Married | 373 | 45.52 |
| Married | 315 | 38.50 |
| Cohabiting | 82 | 9.99 |
| Previously married | 49 | 5.99 |
| **Wealth Index** | | |
| Poorest | 274 | 33.65 |
| Poorer | 249 | 30.36 |
| Middle | 138 | 16.80 |
| Richer | 114 | 13.86 |
| Richest | 44 | 5.33 |
| **Religion** | | |
| Christians | 707 | 86.31 |
| Muslim | 60 | 7.29 |
| Atheists/Other | 52 | 6.40 |
| **Place of residence** | | |
| Urban | 166 | 20.27 |
| Rural | 653 | 79.73 |
| **Currently Working** | | |
| No | 644 | 78.59 |
| Yes | 175 | 21.41 |
| **Number of household members** | | |
| <5 | 311 | 38.00 |
| ≥5 | 508 | 62.00 |
| **Mothers age at first birth** | | |
| 11-15 | 172 | 21.00 |
| 16-19 | 647 | 79.00 |
| **Child's sex** | | |
| Male | 393 | 47.95 |
| Female | 426 | 52.05 |
| **Child's age (months)** | | |
| <6 | 202 | 24.72 |
| 6-23 | 449 | 54.81 |
| ≥24 | 168 | 20.47 |
| **Parity** | | |
| 1 | 649 | 79.24 |
| 2-3 | 170 | 20.76 |
| **Birth order** | | |
| 1 | 719 | 87.82 |

*(Continued)*

**Table 1.** (Continued)

| Variable | Weighted Frequency (N) | Weighted Percentage (%) |
|---|---|---|
| 2-3 | 100 | 12.18 |
| **Fever & cough in the last 2 weeks** | | |
| No | 534 | 65.22 |
| Yes | 285 | 34.78 |
| **Diarrhea in the last 2 weeks** | | |
| No | 594 | 72.52 |
| Yes | 225 | 27.48 |

quintile (n = 276, 33.65%), were unemployed (n = 644, 78.59%), identified as Christians (n = 707, 86.31%), resided in rural areas (n = 653, 79.73%), and lived in households with five or more members (n = 508, 62.00%). Most had their first birth between the ages of 16–19 years (n = 647, 79.00%).

Among the children, slightly more than half were females (n = 426, 52.05%) and were aged 6–23 months (n = 449, 54.81%). Concerning maternal reproductive history, most adolescent mothers were primiparous (n = 649, 79.24%), while 100 (12.18%) reported having two to three children. Additionally, the majority of children had not experienced fever or cough (n = 534, 65.22%) or diarrhoea (n = 594, 72.52%) in the two weeks before the survey.

### Prevalence and distribution of underweight across covariates

The overall prevalence of underweight among children under five years born to adolescent mothers in Kenya was 11.03% (n = 77; 95% CI: 8.65–13.97). The prevalence of underweight was highest among children of mothers with no formal education (n = 9, 18.16%), children of previously married mothers (n = 8, 18.09%), and children of married mothers (n = 39, 14.36%). A higher prevalence was also observed among children aged 24 months and above (n = 24, 16.96%) and those aged 6–23 months (n = 45, 11.79%) (Table 2).

### Factors associated with underweight among under-5 children in Kenya

Table 3 presents the factors associated with underweight among children under five years born to adolescent mothers in Kenya. In the bivariate analysis, maternal education, marital status, and child age were significantly associated with underweight. After adjusting for potential confounders, marital status and the child's age remained significantly associated with underweight status. Children of married adolescent mothers were 2.22 times as likely to be underweight [APR: 2.22; 95% CI: 1.15–4.28; p = 0.018] than children of unmarried adolescent mothers. The prevalence of underweight significantly increased with an increase in child's age. Children aged 24 months and older (APR = 4.35; 95% CI: 1.84–10.30) and those aged 6–23 months (APR = 2.73; 95% CI: 1.22–6.12) were more likely to be underweight than children younger than six months.

### Discussion

This study examined the prevalence and factors associated with underweight among children under five years born to adolescent mothers aged 15–19 years in Kenya. The prevalence of underweight in this population was 11.03% (n = 77/702), indicating that approximately one in nine children of adolescent mothers were underweight. This prevalence is lower than that reported by Wemakor et al. (2018) in Tamale, Ghana, where 29.3% of the children of adolescent mothers were underweight, and lower than the 29.5% reported by Olodu et al. (2019) among children of teenage mothers in southwestern Nigeria [13,26]. This difference may be attributed to variations in the sample size. While these earlier studies were localized and based on smaller or community-level samples, the current study utilized a nationally representative

**Table 2. Prevalence and distribution of underweight across the covariates.**

| Characteristics | Total (N) | Underweight N (%) | Not underweight N (%) | P-value |
|---|---|---|---|---|
| **Overall** | 702 | 11.03 [n=77; 95% CI: 8.65–13.97] | 88.97 [n=625; 95% CI: 86.03–91.35] | |
| **Mother's age** | | | | 0.304 |
| 15-17 | 175 | 24 (13.45) | 152 (86.55) | |
| 18-19 | 527 | 54 (10.22) | 473 (89.78) | |
| **Mother's educational level** | | | | **0.027** |
| No formal education | 47 | 9 (18.16) | 38 (81.84) | |
| Primary | 317 | 42 (13.21) | 275 (86.79) | |
| Secondary or higher | 338 | 27 (7.99) | 311 (92.01) | |
| **Marital status** | | | | 0.060 |
| Not Married | 320 | 25 (7.77) | 295 (92.23) | |
| Married | 270 | 39 (14.36) | 232 (85.64) | |
| Cohabiting | 70 | 6 (8.80) | 64 (91.20) | |
| Previously married | 42 | 8 (18.09) | 34 (81.91) | |
| **Wealth Index** | | | | 0.221 |
| Poorest | 236 | 36 (15.03) | 201 (84.97) | |
| Poorer | 213 | 19 (8.91) | 194 (91.09) | |
| Middle | 118 | 13 (10.91) | 105 (89.09) | |
| Richer | 97 | 9 (9.50) | 88 (90.50) | |
| Richest | 37 | 1 (2.17) | 37 (97.83) | |
| **Religion** | | | | 0.327 |
| Christians | 606 | 64 (10.49) | 542 (89.51) | |
| Muslim | 51 | 9 (17.84) | 42 (82.16) | |
| Atheists/Other | 45 | 5 (10.50) | 40 (89.50) | |
| **Place of residence** | | | | 0.241 |
| Urban | 142 | 10 (7.30) | 132 (92.70) | |
| Rural | 560 | 67 (11.98) | 493 (88.02) | |
| **Currently Working** | | | | 0.709 |
| No | 552 | 62 (11.29) | 489 (88.71) | |
| Yes | 150 | 15 (10.09) | 135 (89.91) | |
| **Number of household members** | | | | 0.749 |
| <5 | 267 | 28 (10.48) | 239 (89.52) | |
| ≥5 | 435 | 49 (11.37) | 386 (88.63) | |
| **Mother's age at first birth** | | | | 0.357 |
| 11-15 | 147 | 19 (13.21) | 128 (86.79) | |
| 16-19 | 555 | 58 (10.45) | 497 (89.55) | |
| **Child's sex** | | | | 0.835 |
| Male | 337 | 38 (11.31) | 299 (88.69) | |
| Female | 365 | 39 (10.77) | 326 (89.23) | |
| **Child's age (months)** | | | | **0.003** |
| <6 | 174 | 8 (4.44) | 166 (95.56) | |
| 6-23 | 385 | 45 (11.79) | 339 (88.21) | |
| ≥24 | 144 | 24 (16.96) | 119 (83.04) | |
| **Parity** | | | | 0.948 |
| 1 | 556 | 62 (11.07) | 495 (88.93) | |
| 2-3 | 146 | 16 (10.89) | 130 (89.11) | |

*(Continued)*

**Table 2.** (Continued)

| Characteristics | Total (N) | Underweight N (%) | Not underweight N (%) | P-value |
|---|---|---|---|---|
| **Birth order** | | | | 0.350 |
| 1 | 617 | 70 (11.39) | 546 (88.61) | |
| 2-3 | 86 | 7 (8.44) | 78 (91.56) | |
| **Fever & cough in the last 2 weeks** | | | | 0.202 |
| No | 458 | 45 (9.76) | 413 (90.24) | |
| Yes | 244 | 33 (13.41) | 211 (86.59) | |
| **Diarrhea in the last 2 weeks** | | | | |
| No | 509 | 56 (11.03) | 453 (88.97) | 0.995 |
| Yes | 193 | 21 (11.02) | 172 (88.98) | |

data from the 2022 Kenya Demographic and Health Survey (KDHS), which enhances the precision and generalizability of the estimates. These findings across diverse contexts reinforce the persistent vulnerability of children born to adolescent mothers to being underweight. The findings correspond with global evidence linking adolescent motherhood to a higher risk of underweight offspring. A systematic review by Welch et al. (2024) documented that children of adolescent mothers have significantly higher odds of being underweight than those of adult mothers across low- and middle-income countries [9]. This finding can be explained by several interrelated factors. Adolescent mothers often face biological immaturity, which results in nutrient competition between the growing mother and fetus, consequently leading to low birth weight and poor postnatal growth trajectories [27]. In addition, socioeconomic disadvantages, such as low income, limited education, and restricted access to health services, further compromise the ability of adolescent mothers to provide optimal nutrition and care for their children [28].

In our multivariable analysis, being in a marital union and the child's age were significantly associated with underweight among children under five years. The risk of being underweight was twice as high among children under five years of adolescent mothers currently in a marital union compared to those of mothers not in a marital union. This finding is consistent with evidence from Sierra Leone, Bangladesh, and India, indicating that the children of married adolescents are more likely to be underweight than those of unmarried adolescents [29–32]. Hossain et al. (2024) and Jama et al. (2018) noted that adolescent motherhood within marital unions is associated with early childbearing, limited autonomy, and reduced decision-making power regarding child nutrition and healthcare [20,33]. Early marriage may also constrain educational attainment and employment opportunities, thereby perpetuating a cycle of poverty and nutritional deprivation. This finding implies that a comprehensive approach addressing both structural and behavioral factors is needed to reduce the prevalence of underweight among children of adolescent mothers in the future. Policies that promote delayed marriage and uninterrupted education for adolescent girls can significantly reduce early childbearing and its associated nutritional effects. Strengthening adolescent-friendly reproductive health and nutrition programs can empower young mothers with knowledge of proper feeding practices and maternal care. Efforts to enhance adolescent mothers' autonomy, such as providing access to income-generating activities and social protection initiatives, can improve their capacity to ensure adequate nutrition for their children.

Furthermore, the child's age was an independent and significant predictor of underweight. The likelihood of being underweight increased progressively with an increase in the child age. Children aged 6–23 months had three times higher odds of being underweight, whereas those aged 24 months and older exhibited more than five times the odds compared with infants under six months. This finding aligns with existing evidence suggesting that the risk of being underweight increases with age, particularly after six months, when complementary feeding is inadequate or delayed [34,35]. The observed protective effect in infants under six months of age is likely attributable to the benefits of exclusive

**Table 3. Factors associated with underweight in children under five years of adolescent mothers.**

| Variable | N (%) | CPR [95% CI] | P-value | APR [95% CI] | P-value |
|---|---|---|---|---|---|
| **Mother's age** | | | | | |
| 18-19 | 24 (13.45) | Ref | | Ref | |
| 15-17 | 54 (10.22) | 1.32 [0.78-2.21] | 0.301 | 1.81 [0.94-3.48] | 0.077 |
| **Education** | | | | | |
| No formal education | 9 (18.16) | Ref | | Ref | |
| Primary | 42 (13.21) | 0.73 [0.43-1.23] | 0.236 | 0.96 [0.53-1.73] | 0.885 |
| Secondary or higher | 27 (7.99) | 0.44 [0.26-0.75] | 0.003 | 0.81 [0.42-1.56] | 0.527 |
| **Marital status** | | | | Ref | |
| Not Married | 25 (7.77) | Ref | | Ref | |
| Married | 39 (14.36) | 1.61 [1.01-2.54] | 0.044 | **2.22 [1.15-4.28]** | **0.018** |
| Cohabiting | 6 (8.80) | 0.78 [0.34-1.82] | 0.564 | 1.35 [0.50-3.66] | 0.551 |
| Previously married | 8 (18.09) | 1.71 [0.79-3.71] | 0.175 | 1.97 [0.75-5.17] | 0.169 |
| **Wealth Index** | | | | | |
| Poorest | 36 (15.03) | Ref | | Ref | |
| Poorer | 19 (8.91) | 0.59 [0.32-1.10] | 0.095 | 0.70 [0.39-1.24] | 0.218 |
| Middle | 13 (10.91) | 0.73 [0.39-1.34] | 0.305 | 0.84 [0.44-1.60] | 0.602 |
| Richer | 9 (9.50) | 0.63 [0.23-1.71] | 0.366 | 0.98 [0.36-2.68] | 0.971 |
| Richest | 1 (2.17) | 0.14 [0.02-1.06] | 0.057 | 0.23 [0.02-2.25] | 0.208 |
| **Religion** | | | | | |
| Christians | 64 (10.49) | Ref | | Ref | |
| Muslim | 9 (17.84) | 1.70 [0.86-3.37] | 0.128 | 1.24 [0.61-2.51] | 0.555 |
| Atheists/Other | 5 (10.50) | 1.00 [0.38-2.61] | 0.999 | 1.12 [0.45-2.84] | 0.804 |
| Place of residence | | | | | |
| Urban | 10 (7.30) | Ref | | Ref | |
| Rural | 67 (11.98) | 1.64 [0.70-3.85] | 0.255 | 1.39 [0.55-3.51] | 0.486 |
| **Currently Working** | | | | | |
| No | 62 (11.29) | Ref | | Ref | |
| Yes | 15 (10.09) | 0.89 [0.49-1.62] | 0.710 | 0.78 [0.44-1.38] | 0.396 |
| **Number of household members** | | | | | |
| <5 | 28 (10.48) | Ref | | Ref | |
| ≥5 | 49 (11.37) | 1.09 [0.66-1.79] | 0.749 | 1.29 [0.75-2.20] | 0.356 |
| **Mothers age at first birth** | | | | | |
| 11-15 | 19 (13.21) | 1.26 [0.77-2.08] | 0.355 | 0.80 [0.43-1.46] | 0.461 |
| 16-19 | 58 (10.45) | Ref | | Ref | |
| Child's sex | | | | | |
| Male | 38 (11.31) | Ref | | Ref | |
| Female | 39 (10.77) | 0.95 [0.60-1.52] | 0.835 | 0.94 [0.62-1.42] | 0.756 |
| **Child's age (months)** | | | | | |
| <6 | 8 (4.44) | Ref | | | |
| 6-23 | 45 (11.79) | 2.65 [1.18-5.96] | 0.018 | **2.73 [1.22-6.12]** | **0.015** |
| ≥24 | 24 (16.96) | 3.82 [1.65-8.82] | 0.002 | **4.35 [1.84-10.30]** | **0.001** |
| **Parity** | | | | | |
| 1 | 62 (11.07) | Ref | | Ref | |
| 2-3 | 16 (10.89) | 0.98 [0.60-1.62] | 0.948 | 0.62 [0.25-1.55] | 0.315 |
| **Birth order** | | | | | |
| 1 | 70 (11.39) | Ref | | Ref | |

*(Continued)*

**Table 3.**  (Continued)

| Variable | N (%) | CPR [95% CI] | P-value | APR [95% CI] | P-value |
|---|---|---|---|---|---|
| 2-3 | 7 (8.44) | 0.74 [0.39-1.41] | 0.357 | 0.99 [0.38-2.56] | 0.979 |
| **Fever & cough in the last 2 weeks** | | | | | |
| No | 45 (9.76) | Ref | | Ref | |
| Yes | 33 (13.41) | 1.37 [0.84-2.23] | 0.201 | 1.41 [0.88-2.27] | 0.156 |
| **Diarrhea in the last 2 weeks** | | | | | |
| No | 56 (11.03) | Ref | | Ref | |
| Yes | 21 (11.02) | 1.00 [0.60-1.66] | 0.995 | 0.89 [0.52-1.55] | 0.687 |

CPR = crude prevalence ratio; APR = adjusted prevalence ratio; CI = confidence interval; Ref = Reference category.

breastfeeding, which is more prevalent during this early period of life. In contrast, the initiation of complementary feeding at around six months, is a critical period that can impact a child's nutritional status and development. During this phase, the risk of underweight increases due to heightened nutritional needs that are often unmet [36]. This elevated risk is often associated with suboptimal feeding practices, including poor dietary quality, infrequent feeding, and limited food diversity. Another contributing factor is the inadequate knowledge of infant and young child feeding practices among adolescent mothers [37]. Consequently, children born to adolescent mothers aged six months and above may lack adequate nutrients necessary for growth and development.

In Kenya, several studies have highlighted suboptimal complementary feeding practices, particularly among young mothers with low income or limited knowledge of child nutrition [38,39]. Adolescent mothers often encounter significant challenges in initiating appropriate complementary feeding at the recommended age of six months. Barriers such as household food insecurity, limited social support, and inadequate knowledge or misconceptions regarding infants and young children can hinder timely and adequate dietary transitions. These constraints may result in suboptimal energy and nutrient intake during a critical period of rapid growth and development, thereby increasing the risk of underweight in older infants and toddlers. Moreover, this finding highlights the cumulative burden of underweight over time, as the compounded effects of prolonged dietary inadequacy, recurrent infections, and restricted access to healthcare services become more pronounced with increasing age. These findings highlight the need for public health interventions to support adolescent mothers in adopting appropriate complementary feeding practices and preventing and treating infections in children. In line with established public health and nutritional guidelines, interventions should prioritise enhancing maternal knowledge, promoting dietary diversity, and improving access to nutritionally adequate foods for their children.

## Strengths and limitations

The main strength of this study lies in the use of nationally representative data from the 2022 Kenya Demographic and Health Survey, which enhances the generalizability of the findings to children born to adolescent mothers aged 15–19 years. Secondly, the data collection procedures were standardized and replicable, contributing to the reliability of the study. Additionally, children's nutritional status was assessed using the weight-for-age Z-score, a validated measure that strengthens the reliability of the findings regarding underweight prevalence. However, the cross-sectional design of the survey limits the ability to draw causal inferences on the relationship between independent variables and underweight. Moreover, important factors such as maternal nutritional status during pregnancy, household food availability, and breast-feeding or complementary feeding practices were not included in the analysis. These unmeasured confounders may have introduced bias into the observed associations and warrant further investigation that accounts for them. Lastly, multicollinearity was assessed using VIF; however, VIF has limitations when applied to models including categorical predictors

and non-linear regression models. Therefore, residual multicollinearity among categorical variables may not have been fully captured, and coefficient estimates should be interpreted with caution.

## Conclusion

The prevalence of underweight among children under five years of adolescent mothers aged 15–19 years in Kenya was 11.03%. Marital status and the child's age were significantly associated with underweight. Children of married adolescent mothers were over 2.22 times as likely to be underweight compared to children of unmarried adolescent mothers. The prevalence of underweight also increased with an increase in the age of the child, with children aged 6–23 months and those aged 24 months and older being more likely to be underweight than children younger than six months. Future research should investigate complementary feeding practices among adolescent mothers to understand how feeding timing, dietary diversity, frequency, and cultural practices influence the risk of undernutrition in children under five years.

## Acknowledgments

We acknowledge the MEASURE DHS project for providing access to the Kenya Demographic and Health Survey 2022 dataset, which made this analysis possible.

## Author contributions

**Conceptualization:** Alfred Omutoj, Prince Tsekpetse.

**Formal analysis:** Henry Prosper Dade, Prince Tsekpetse.

**Methodology:** Samuel Salu.

**Supervision:** Prince Tsekpetse.

**Writing – original draft:** Alfred Omutoj, Henry Prosper Dade, Annitah Kagali, Samuel Salu, David Mensah Otoo, Betty Oloo, Prince Tsekpetse.

**Writing – review & editing:** Alfred Omutoj, Henry Prosper Dade, Annitah Kagali, Samuel Salu, David Mensah Otoo, Betty Oloo, Prince Tsekpetse.

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
