## [Decision Letter · Decision Letter 0]

16 Oct 2025

Dear Dr. Dade,

Thank you for submitting your manuscript to PLOS ONE. After careful consideration, we feel that it has merit but does not fully meet PLOS ONE’s publication criteria as it currently stands. Therefore, we invite you to submit a revised version of the manuscript that addresses the points raised during the review process.

Revise the manuscript as per review comments.

We look forward to receiving your revised manuscript.

Kind regards,

Md. Moyazzem Hossain, PhD

Academic Editor

PLOS ONE

Journal Requirements:

Reviewers' comments:

Reviewer's Responses to Questions

**Comments to the Author**

1. Is the manuscript technically sound, and do the data support the conclusions?

Reviewer #1: Yes

Reviewer #2: No

Reviewer #3: Yes

2. Has the statistical analysis been performed appropriately and rigorously?

Reviewer #1: Yes

Reviewer #2: No

Reviewer #3: Yes

3. Have the authors made all data underlying the findings in their manuscript fully available?

Reviewer #1: Yes

Reviewer #2: Yes

Reviewer #3: Yes

4. Is the manuscript presented in an intelligible fashion and written in standard English?

Reviewer #1: Yes

Reviewer #2: Yes

Reviewer #3: Yes

Reviewer #1: 1. Technical quality and data support for conclusions

The manuscript is well-structured and follows standard scientific reporting. The analysis is based on nationally representative KDHS 2022 data with 819 children, and the main findings (11.03% prevalence of underweight; significant predictors being younger maternal age 15–17 years and older child age) are consistent with the evidence. The conclusions are clearly supported by the data.

2. Statistical analysis

The statistical procedures were appropriately applied using Stata 17 with survey weights and complex design adjustments. Bivariate and multivariable logistic regression models were presented with ORs, AORs, and 95% CIs. Multicollinearity was checked (mean VIF 1.50), ensuring model validity. The analysis is rigorous overall, with only minor inconsistencies in table formatting.

3. Data availability

The data are publicly accessible through the DHS Program, and key results are transparently presented in detailed tables. Although no supplementary files with raw outputs are provided, the study meets journal standards for data transparency and reproducibility.

4. Language and presentation

The manuscript is written in clear academic English and adheres to STROBE guidelines. It is easy to follow, though minor typographical errors are present and could be corrected with light proofreading. Overall, the presentation is sound and meets international publication standards.

Reviewer #2: This study examined the prevalence and factors associated with underweight among Kenyan children under five years of age born to adolescent mothers, using nationally representative data from the 2022 Kenya Demographic and Health Survey (KDHS). But unfortunately, the study did not follow the proper analysis guidelines. The methods section is confusing and incomplete. This manuscript needs more improvement and clear justification. My specific observations in below:

Comment 1: In the abstract, the author mentioned, “This study examined the prevalence and factors associated with underweight”. I think, author should add “prevalence” in the title.

Comment 2: The method section will be more understandable if the author creates a short flow chart for study design and data processing, including the inclusion and exclusion criteria, the number of weighted and unweighted observations, and missing values. It is already noticeable that the methods section didn’t specify the number of samples taken before and after weighting. Additionally, the number of households or cities or villages that were covered in the study isn’t mentioned.

Comment 3: The author needs to use a reference to the prior study mentioned for using VIF, in lines 32-33 on page 6. Also mentioned “The mean VIF was calculated at 1.50.”, needs more explanation or no need to keep this line in this section.

Comment 4: In page 6, line 36, “Variables with a p-value less than 0.2 were included in the multivariable analysis.” But why? This line needs justification.

Comment 5: The use of percentages in the manuscript is inconsistent. The authors use single decimal digits in some places and two decimal places elsewhere. It should be made consistent.

Comment 6: In Table 2, the value of N doesn’t sum up to 819 for all variables, such as mothers' education level and mothers' age, as N=820. Check all the frequencies and percentages of each table and fix the error in the manuscript carefully.

Comment 7: In tables 3 and 4, the author didn’t mention the reference category for the response variable.

Comment 8: Did the author consider sampling weight for modelling and other analyses? If not, please explain the reasons. I think it should be used in DHS-related nationwide survey research.

Comment 9: Table 4 has the same COR values as Table 3; the author could present them in the same table. I cannot see any significance in Table 3 output, especially where this information also exists in Table 4. But, I wonder how the author could say the logistic regression is a bivariate analysis? I think authors should discuss the analysis plan with a professional statistician.

Comment 10: Of all the independent variables there shows only two variables are significant! But why? Additionally, as you already revealed, 80% of the variables where not have any association between the outcome variable and the independent variable according to the chi-square test. So how could the author think about the significant influencing factor for the outcome variable? Need proper justification.

Reviewer #3: Dear authors

I have the following comments , I hope to be answered for improving this article

The present title is “Factors associated with underweight among children under five born to adolescent mothers in Kenya: Evidence from the 2022 Kenya Demographic and Health Surve”

Introduction

1. Shift reference no. (5) to the end of the sentence

2. Both types of references’ citation were used (Harvard and Vancouver styles) for the reference number (9,11).

3. What about the management guidelines for the treatment of cases of underweight , stunting , and wasting in Ghana.

Methods

1. Add more details on sampling procedure “stratified, two-stage cluster sampling design”

2. What about the nutritional practices , feeding type, any supplementary foods that the children received

3. The criteria used for determining wealth index

4. I think the unmarried mother and cohabiting is the same status of the marital status , if not , please clarify.

5.

Results

1. The results in the text , it is better to be highlighted in percentages only.

2. In line 56 , Put dash(-) between the two CI limits instead of comma [CI: 8.65% -13.97%]

Discussion

1. This sentence “ sub-Saharan Africa continues to report over 100 births per 1,000 adolescent girls, with approximately 6.1 million births annually among those aged 15 to 19 years (10)”It was mentioned in introduction section.

**Do you want your identity to be public for this peer review?** For information about this choice, including consent withdrawal, please see our Privacy Policy

Reviewer #1: No

Reviewer #2: **Yes:** Moinur Rahman

Reviewer #3: **Yes:** Masood Abdulkareem Abdulrahman

---

## [Author Response · Author response to Decision Letter 1]

23 Oct 2025

REVIEWER #1:

GENERAL COMMENTS: We sincerely thank Reviewer 1 for their constructive comments and positive evaluation. We have carefully proofread the manuscript to correct the minor typographical issues noted. These included standardizing decimal places in percentages and ensuring consistency in formatting across tables and references.

REVIEWER #2:

COMMENT 1: In the abstract, the author mentioned, “This study examined the prevalence and factors associated with underweight”. I think the author should add “prevalence” in the title.

Response: Thank you for this suggestion. We agree with your recommendation and have revised the title accordingly. Revised Title: Prevalence and factors associated with underweight among children under five born to adolescent mothers in Kenya: Evidence from the 2022 Kenya Demographic and Health Survey

COMMENT 2: The method section will be more understandable if the author creates a short flow chart for study design and data processing, including the inclusion and exclusion criteria, the number of weighted and unweighted observations, and missing values.

Response: Thank you for the helpful suggestion. In response, we have developed and included a flowchart (Figure 1) in the methods section summarising study design, inclusion/exclusion criteria, and sample derivation.

COMMENT 3: Use a reference for the VIF justification and explain the meaning of VIF.

Response: We retained the sentence and expanded the explanation with a supporting citation:

Revised Section in Methods: "Multicollinearity was assessed using the variance inflation factor (VIF) to determine the extent of the correlation among the independent variables. The VIF helps identify predictors that may distort regression estimates, as high values indicate multicollinearity and unstable coefficients (O’Brien, 2007)."

COMMENT 4: Justify why a p-value threshold of 0.2 was used for including variables in multivariable analysis.

Response: The methods section now states: "A bivariate model was first fitted, and independent variables with a p-value less than 0.20 were included in the multivariable analysis to avoid excluding potentially important predictors. This threshold allows the inclusion of variables that may not reach conventional levels of significance in the bivariate analysis but could be significant in the multivariable context due to confounding or interaction (Hosmer et al., 2013)."

COMMENT 5: Fix inconsistent use of decimal places in percentages.

Response: All percentages in text and tables have been standardized to two decimal place.

COMMENT 6: N value inconsistencies in Table 2.

Response: These discrepancies have been corrected.

COMMENT 7: Reference categories missing in regression tables.

Response: Reference categories are now explicitly labelled in all relevant tables.

COMMENT 8: Confirm whether sampling weights were used in modelling.

Response: The methods section has been updated to confirm that all estimates, including regression models, accounted for survey design using the "svy" command in Stata.

COMMENT 9: Redundancy between Tables 3 and 4.

Response: We merged the bivariate and multivariable results into a single comprehensive table to avoid redundancy. This change is reflected in the Results section.

COMMENT 10: Of all the independent variables there shows only two variables are significant; justification needed. Also, bivariate chi-square test showed 80% non-significance.

Response: Thank you for your observation. We provide the following justification regarding our model-building strategy and interpretation of results:

Many variables did not show statistical significance in the initial bivariate analysis, as presented in Tables 2 and 3 (CPR column). This is why we used a more inclusive p-value threshold of 0.20 for variable selection in the multivariable model, a common epidemiological practice to avoid excluding important confounders. Additionally, we retained key variables known from prior literature on undernutrition regardless of their bivariate significance to ensure a well-adjusted model. The multivariable analysis aims to identify independent predictors by adjusting for confounders. It is expected that some variables significant in bivariate analysis become non-significant after adjustment. For example, maternal secondary education was significant in the crude analysis but not in the adjusted model, suggesting its effect is mediated by other factors. Therefore, the finding that only marital status and child’s age remain significant demonstrates these are the strongest independent predictors of underweight, reinforcing rather than weakening our conclusions. This approach ensures robust identification of key factors affecting underweight among children of adolescent mothers.

REVIEWER #3:

COMMENT 1: Shift reference (5) to the end of the sentence.

Response: This has been corrected in the Introduction section.

COMMENT 2: Citation style inconsistency.

Response: All in-text citations and references were revised to comply with the Vancouver style used by PLOS ONE. We used EndNote Referencing software to ensure consistency.

COMMENT 3: Add mention of management guidelines for underweight.

Response: Response: Thank you for this valuable suggestion. We agree that linking our study's implications to established guidelines is crucial.

We have revised the discussion section to explicitly state that our recommendations are directly informed by and aligned with existing public health and nutritional guidelines for managing undernutrition. This clarification strengthens the basis for our proposed interventions.

The revised text now reads: "These findings highlight the need for public health interventions to support adolescent mothers in adopting appropriate complementary feeding practices... In line with established public health and nutritional guidelines, interventions should prioritise enhancing maternal knowledge, promoting dietary diversity, and improving access to nutritionally adequate foods for their children.

COMMENT 4: Expand explanation of sampling procedure.

Response: We added details on KDHS's two-stage stratified sampling method in the Methods section.

Comment 5: Discuss feeding practices.

Response: We acknowledged that the dataset lacked detailed feeding data for this subgroup and stated this limitation section of the manuscript.

COMMENT 6: Clarify wealth index construction.

Response: We noted that the wealth index was derived using PCA, following DHS standard methodology.

COMMENT 7: Clarify marital status definitions.

Response: Thank you for your comment. Our study used secondary data from the 2022 Kenya Demographic and Health Survey, where marital status was classified into standard categories as follows: not married (including single or never married), married (currently in a legal marital union), cohabiting (living together without formal marriage), and previously married (divorced, separated, or widowed).

COMMENT 8: Present results as percentages.

Response: The Results section was revised to emphasise percentage reporting over absolute counts.

COMMENT 9: Use dash instead of comma for CI.

Response: Fixed throughout manuscript (e.g., 95% CI: 8.65% - 13.97%).

COMMENT 10: The sentence “sub-Saharan Africa continues to report over 100 births per 1,000 adolescent girls, with approximately 6.1 million births annually among those aged 15 to 19 years (10)” was duplicated in the Discussion.

Response: Thank you for this observation. The repeated sentence has been removed from the Discussion section to avoid redundancy. The original statement remains appropriately cited in the Introduction.

---

## [Decision Letter · Decision Letter 1]

1 Jan 2026

Dear Dr. Dade,

Thank you for submitting your manuscript to PLOS ONE. After careful consideration, we feel that it has merit but does not fully meet PLOS ONE’s publication criteria as it currently stands. Therefore, we invite you to submit a revised version of the manuscript that addresses the points raised during the review process.

We look forward to receiving your revised manuscript.

Kind regards,

Md. Moyazzem Hossain, PhD

Academic Editor

PLOS One

Journal Requirements:

Reviewers' comments:

Reviewer's Responses to Questions

**Comments to the Author**

Reviewer #2: All comments have been addressed

2. Is the manuscript technically sound, and do the data support the conclusions?

Reviewer #2: Yes

3. Has the statistical analysis been performed appropriately and rigorously?

Reviewer #2: Yes

4. Have the authors made all data underlying the findings in their manuscript fully available?

Reviewer #2: Yes

5. Is the manuscript presented in an intelligible fashion and written in standard English?

Reviewer #2: Yes

Reviewer #2: Comment 1: Though the author has newly added a flow chart, which is good for sample selection and exclusion criteria, it is not well-defined. Please use “Yes”, “No” and directional signs in the proper place from 1st to the last step.

Comment 2: In table 1, it mentions “Characteristics of the study participants (Unweighted N=819)” but in the table header uses weighted frequency. Please check it carefully.

Comment 3: I wonder how the author could calculate the VIF value for a categorical variable and why they described “as high values indicate multicollinearity and unstable coefficients” instead of the standard cutoff value of VIF. The author should acknowledge this issue in the limitations section.

**Do you want your identity to be public for this peer review?** For information about this choice, including consent withdrawal, please see our Privacy Policy

Reviewer #2: **Yes:** Moinur Rahman

---

## [Author Response · Author response to Decision Letter 2]

3 Jan 2026

Dear Editor and Reviewer,

We sincerely thank the Editor and Reviewer #2 for their careful evaluation of our manuscript entitled “Prevalence and factors associated with underweight among children under five born to adolescent mothers: Evidence from the 2022 Kenya Demographic and Health Survey.” We appreciate the constructive comments, which have helped to improve the clarity, methodological transparency, and overall quality of the manuscript.

Below, we provide a point-by-point response to each comment raised by Reviewer #2. All revisions have been incorporated into the revised manuscript, with changes highlighted in the tracked-changes version.

Reviewer #2 – Comment 1

“Though the author has newly added a flow chart, which is good for sample selection and exclusion criteria, it is not well-defined. Please use ‘Yes’, ‘No’ and directional signs in the proper place from 1st to the last step.”

Response:

We thank the reviewer for this helpful suggestion. The flowchart has been revised to improve clarity and logical flow. Specifically, we have:

• Reorganised the sequence from the first to the final step of sample selection,

• Added clear directional arrows to guide the reader,

• Explicitly indicated inclusion and exclusion steps using consistent labels and directional flow.

Reviewer #2 – Comment 2

“In table 1, it mentions ‘Characteristics of the study participants (Unweighted N=819)’ but in the table header uses weighted frequency. Please check it carefully.”

Response:

Thank you for highlighting this inconsistency. We have carefully reviewed Table 1 and confirm that the frequencies presented are weighted, in line with DHS analytical recommendations. The table title has now been corrected to indicate a weighted sample size (Weighted N = 819) to ensure consistency between the table heading and the reported statistics.

Reviewer #2 – Comment 3

“I wonder how the author could calculate the VIF value for a categorical variable and why they described ‘as high values indicate multicollinearity and unstable coefficients’ instead of the standard cutoff value of VIF. The author should acknowledge this issue in the limitations section.”

Response:

We thank the reviewer for this important methodological observation. We have addressed this concern in two ways:

• Specified standard cutoff values for VIF interpretation (VIF > 5 indicating moderate multicollinearity and VIF > 10 indicating serious multicollinearity).

• Explicitly acknowledged the limitation of applying VIF in models including categorical predictors and non-linear regression in the Strengths and Limitations section, noting that residual multicollinearity may not be fully captured and that coefficient estimates should be interpreted with caution.

---

## [Decision Letter · Decision Letter 2]

26 Jan 2026

Prevalence and factors associated with underweight among children under five born to adolescent mothers: Evidence from the 2022 Kenya Demographic and Health Survey

PONE-D-25-46014R2

Dear Dr. Dade,

We’re pleased to inform you that your manuscript has been judged scientifically suitable for publication and will be formally accepted for publication once it meets all outstanding technical requirements.

Kind regards,

Md. Moyazzem Hossain, PhD

Academic Editor

PLOS One

Additional Editor Comments (optional):

Reviewers' comments:

Reviewer's Responses to Questions

**Comments to the Author**

Reviewer #2: All comments have been addressed

2. Is the manuscript technically sound, and do the data support the conclusions?

Reviewer #2: Yes

3. Has the statistical analysis been performed appropriately and rigorously?

Reviewer #2: Yes

4. Have the authors made all data underlying the findings in their manuscript fully available?

Reviewer #2: Yes

5. Is the manuscript presented in an intelligible fashion and written in standard English?

Reviewer #2: Yes

Reviewer #2: All comments have been correctly specified. According to my perspective, this paper is methodologically sound and publishable. Thanks

**Do you want your identity to be public for this peer review?** For information about this choice, including consent withdrawal, please see our Privacy Policy

Reviewer #2: **Yes:** Moinur Rahman

---

## [Editor Report · Acceptance letter]

PONE-D-25-46014R2

PLOS One

Dear Dr. Dade,

I'm pleased to inform you that your manuscript has been deemed suitable for publication in PLOS One. Congratulations! Your manuscript is now being handed over to our production team.

Kind regards,

on behalf of

Professor Md. Moyazzem Hossain

Academic Editor

PLOS One